# Surgical debulking of podoconiosis nodules and its impact on quality of life in Ethiopia

**Wendemagegn Enbiale**[1,2]*, **Kristien Verdonck**[3], **Melesse Gebeyehu**[1], **Johan van Griensven**[3], **Henry J. C. de Vries**[2,4]

**1** Bahir Dar University, college of Medicine and Health sciences, Bahir Dar, Ethiopia, **2** Amsterdam UMC, University of Amsterdam, department of dermatology, Amsterdam Institute for Infection and Immunity (AII), location Academic Medical Centre, Amsterdam, The Netherlands, **3** Institute of Tropical Medicine, Antwerp, Belgium, **4** STI outpatient clinic, Department of Infectious Diseases, Public Health Service Amsterdam, The Netherlands

* wendemagegnenbiale@gmail.com

**Data Availability Statement:** All relevant data are within the manuscript.

## Abstract

### Background

In Ethiopia, severe lymphedema and acute dermato-lymphangio-adenitis (ADLA) of the legs as a consequence of podoconiosis affects approximately 1.5 million people. In some this condition may lead to woody-hard fibrotic nodules, which are resistant to conventional treatment. We present a series of patients who underwent surgical nodulectomy in a resource-limited setting and their outcome.

### Methods

In two teaching hospitals, we offered surgical nodulectomies under local anaesthesia to patients with persisting significant fibrotic nodules due to podoconiosis. Excisions after nodulectomy were left to heal by secondary intention with compression bandaging. As outcome, we recorded time to re-epithelialization after surgery, change in number of ADLA episodes, change in quality of life measured with the Dermatology Quality of Live Index (DQLI) questionnaire, and recurrence rate one year after surgery.

### Results

37 nodulectomy operations were performed on 21 patients. All wounds re-reepithelialised within 21 days (range 17–42). 4 patients developed clinically relevant wound infections. The DLQI values were significantly better six months after surgery than before surgery (P<0.0001). Also the number of ADLA episodes per three months was significantly lower six months after surgery than before surgery (P<0.0001).

### Conclusion

Nodulectomy in podoconiosis patients leads to a significant improvement in the quality of life with no serious complications, and we recommend this to be a standard procedure in resource-poor settings.

**Funding:** The author(s) received no specific funding for this work.

**Competing interests:** The authors have declared that no competing interests exist.

## Author summary

Podoconiosis is a disease that causes swelling of the lower legs and tissue lumps (nodules) of the feet. The condition is physically disabling with significant psycho-social impact. It is caused by the destruction of the lymphatic system, that is critical in the transportation of body fluids. The disease is said to be caused by long term barefooted exposure to red-clay soil. Podoconiosis affects more than four million subsistent farmers in the tropics. Ethiopia is one of the endemic countries with more than 1.5 million affected people.

Early-stage podoconiosis morbidity management consists of foot hygiene, regular inspection and care for traumatic abrasions, use of moisturizer, daily exercise, compression with elastic bandages, and last but not least shoe wearing. In the late disease stage, irreversible swelling and woody-hard fibrous nodules arise. As a result, patients can no longer wear shoes and suffer from disabling bacterial infection. Both complications aggravate lymphatic destruction. Moreover, late-stage podoconiosis is unresponsive to the aforementioned therapy options. Surgical intervention removing the nodules (nodulectomy) can potentially alleviate late-stage disease.

Here, we describe the clinical outcome of nodulectomy and its effect on the quality of life in 21 late stage podoconiosis patients, in a before to 12 months after the procedure comparison. We demonstrate that nodulectomy is safe, allows patients to wear shoes again, reduces the number of bacterial infections, and significantly improves their quality of life. Nodulectomy is a safe and effective procedure for podoconiosis patients. We recommend nodulectomy as the preferred therapy option for late-stage podoconiosis complicated by nodules.

## Background

Podoconiosis, or non-filarial elephantiasis is one of the causes of lymphedema. The disease affects barefoot subsistence farmers exposed to soil originating from volcanic rock in wet rural highland areas, in a population with genetic susceptibility [1–5]. Globally, podoconiosis affects an estimated 4 million people in about 17 countries, mostly in areas of tropical Africa[6].

In Ethiopia, about one third of the districts in the country are endemic for podoconiosis, with more than 35 million people at risk[7]. Within these endemic areas, population-based surveys have produced prevalence estimates of 5–10%. Overall, podoconiosis is thought to affect about 1.5 million people in the country [7]. The disease is physically disabling and has severe economic consequences: productivity losses per patient amount to 45% and sum up to huge economic losses at country level [8,9]. Stigmatization of people with podoconiosis is marked: some patients are excluded from social gatherings, school, churches and mosques, and are barred from marriage with unaffected individuals [10].

On clinical examination, these patients often have a dry, hyperkeratotic skin with deep cracks in the heels and between the toes. Many patients go on to develop gross oedema and fibrotic nodular and tumorous masses [11,12]. Untreated patients suffer episodes of acute dermato-lymphangio-adenitis(ADLA), with an associated risk of morbidity and mortality[13]. Further more, clinical observation showed that recurrent ADLA results in exacerbation of the lymphedema. The severity of the disease can be classified using a locally modified, simple, five-stage method based on the level of oedema, presence of nodules/tumours, and fixation of the joint [14].

Edema may occur in two forms: pitting or non-pitting (woody hard and fibrotic). The non-pitting form is especially troublesome for the patient because it is associated with multiple, fibrotic skin nodules that cause pain on shoe wearing, and that are easily traumatized leading to ulceration. Moreover, in the non-pitting form of edema, the skin folds are prone to maceration and fungal and bacterial infection [12]. This results in a very low quality of life for patients with nodules [15].

Conventional therapy consists of foot hygiene, regular inspection and care for traumatic abrasions, use of moisturizer, daily exercise, leg elevation, shoe wearing and compression with short-stretched elastic bandages. Although rigorous evidence is lacking, such conventional preventive therapy seems to improve the lymphedema and stop the recurrent ADLA for most patients with mild forms of podoconiosis (stage I and II lymphedema)[13,16]. By contrast, patients with non-pitting oedema and woody-hard fibrous nodules on foot and leg tend to have a poor response to conventional therapy. The oedema and nodules do not respond to compression therapy and the patients have frequently recurring episodes of ADLA. Removal of the nodules in these patients could allow them to make better use of the conventional therapy and decrease the episodes of ADLA[17].

Previously, we reported on11 patients with podoconiosis nodules who underwent nodulectomysince 2010. They showed satisfactory healing rates and a limited number ofseriouscomplications[17]. A PubMed and gray literature search for surgical debulkingornodulectomy for podoconiosis patient returnedonly one additional reference[11].

The aim of the current report is to describe a series of patients with lymphedema and nodules due to podoconiosis who underwent nodulectomy in a referral hospital in the Amhara region of Ethiopia between October 2015 and September 2019. We document time to re-epithelialisation after nodulectomy, change in number of ADLA episodes, change in quality of life, and recurrence of nodules one year after surgery.

## Material and methods

### Ethics statement

Permission to conduct the study was obtained from the teaching hospitals. Local ethics approval was received from the Bahir Dar University College of Medicine and Health Science Ethics review board, reference number 0250/2019, Bahir Dar, Ethiopia. Since nodulectomy and the follow-up management are considered routine care and we used retrospective data without patient identifiers, informed patient consent was not deemed necessary. For the publication of patient photos, oral informed consent was obtained, and any form of patient identification was avoided.

### Study design, setting and population

This is a retrospective study reporting on a series of 21 patients with podoconiosis who underwent nodulectomy in one of two teaching hospitals in Bahir Dar, Ethiopia between January 2015 and December 2019.

Ethiopia is a located in East Africa and has a population of about 103 million. The country is a federal state with nine regional and two special administrations [18], and is challenged with limited health service coverage (39% of the population) [14]. The Amhara regional state lies in the north west of the country and has approximately 21 million inhabitants, 85% of whom live in rural settings [18]. This region has a podoconiosis prevalence of 3.7% and shoe-wearing is strongly advised and promoted as primary and secondary prevention [19,20]. There are two teaching hospitals in Bahir Dar, the capital city of the region: The Felege Hiwot Specialized Hospital and Tibebe Ghion Specialized Hospital. These two hospitals each have about

1500outpatient consultations daily. They both have a specialized dermatology clinic attending cases referred from district hospitals in the region and neighbouring areas. Among other services, the dermatology clinics provide surgical nodulectomy for patients with podoconiosis. Patient with podoconiosis and nodules who have been given nodulectomy service ware recruited from the general dermatology clinics.

### Diagnosis of podoconiosis

In the referral hospitals, we used locally developed clinical diagnostic criteria for podoconiosis combining major and minor criteria (see Box 1). All participants were examined by consultant dermatologists using these criteria.

### Nodulectomy procedure and patient follow-up

Nodulectomy is a surgical reduction of redundant skin/swellings on the foots and lower leg. Nodulectomy was performed under local anaesthesia in a surgical theatre. Fibrotic nodules and tumours were excised with the narrowest possible base. Redundant skin was removed and haemostasis achieved using simple gauze compression and artery forceps. The excision surface area was measured and the weight of the removed tissue was noted. Subsequently, the surgical field was cleaned with normal saline and dressed with sterile gauze and compression bandaging to facilitate healing. The wound was left to heal by secondary intention. Antibiotics and analgesics were given at the surgeon's discretion. In patients with extensive nodules, nodulectomy was temporised in multiple subsequent procedures. In addition, patients were instructed on foot hygiene, skin care, bandaging (for those with water bag lymphedema), exercises, and use of socks and shoes. Patient follow-up took place two, four and eight weeks after nodulectomy, and subsequently every three months for one year. Digital photos were taken before and immediately after nodulectomy and on each follow-up visit.

### Endpoints

We documented the following endpoints: time until re-epitelization, presence of complications (e.g. occurrence of wound infection and non-healing ulcer), number of episodes of ADLA over a period of three months, Dermatology Quality of Life Index (DQLI) at two months and at six months after surgery compared with DQLI at enrolment (before surgery), and recurrence of nodules in the course of the first year after surgery.

An ADLA episode was defined as two or more of the following symptoms or signs: redness, pain, or swelling of the leg or foot, with or without the presence of fever or chills [21]. The number of ADLA episodes was recorded on each follow-up visit.

The DLQI is a widely used dermatology-specific questionnaire consisting of 10 questions concerning health-related quality of life for people with a skin condition[22,23]. In this Ethiopian setting, we used an Amharic version (local language) of the questionnaire [23]. We performed a DLQI assessment prior to the first nodulectomy and 2and 6 months after the last surgical nodulectomy. The DLQI score can be interpreted as follows: 0–1, 'no effect at all'; 2–5, 'small effect'; 6–10, 'moderate effect'; 11–20, 'very large effect'; 21–30, 'extremely large effect' on quality of life [24–26]. Based on an overview on the use of DLQI in various dermatological diseases, we considered that a decrease in DLQI score of 5 points after surgery indicated a meaningful improvement in quality of life[27].

Box 1. Clinical diagnostic criteria for podoconiosis

### Clinical definition

Podoconiosis is a non-communicable geochemically induced tropical form of lymph-edema resulting in bilateral swelling of the lower legs (below-knee) and impaired skin barrier function, acquired through prolonged barefoot exposure to red clay soils of volcanic origin [8].

### Diagnostic criteria

*Major criteria*

1. Below-knee lymphedema

2. Residence in endemic area during development of lymphedema

3. Barefooted walking for prolonged periods of time(more than 10 years)

4. Mossy foot in slipper pattern

*Minor criteria*

1. Nodules

2. Toe fusion

3. Both feet affected

4. Positive Stemmer's sign test (i.e., the thickened fold of skin at base of the second toe that can be pinched and lifted. This test is positive when the skin cannot be lifted, showing presence of lymphedema)

5. Burning sensation on the feet

6. Family history of the same illness

7. Problem started in the first three decades of life

**Diagnosis**

1. *Definitive podoconiosis*: 3 major or 2 major plus 2 minor or 1 major plus 5 minor criteria

2. *Probable podoconiosis*: 2 major or 1 major plus 2 minor or 5 minor criteria

## Data collection and analysis

Patient data were extracted directly from the patient file and health information management forms, and entered into an Excel data base. The following information was extracted: demographic characteristics, start date and duration of edemaand nodules, staging, history of ADLA episodes, the surgical procedure, antibiotic use and wound care during follow-up, signs of infection, duration of re-epithelization, and recurrence of nodules. To prevent data entry errors, the data was subsequently cross-checked the patient files and Excel database systematically.

We computed descriptive statistics (proportion, median, range, and interquartile range) to summarise patient characteristics, organised per patient and per surgery episode. We compared DLQI score before surgery, two months and six month after surgery and number of ADLA episodes before surgery and three months after surgery (paired comparison) and used the exact Wilcoxon signed rank test to assess statistical significance (one-sided test with a significance level of 0.05). The null hypothesis was that the values after surgery were equal or worse than before surgery. R software was used for data analysis and visualisation.

## Results

### Demographic and clinical characteristics of the patient population

Between October 2015andSeptember 2019, 32 new patients underwent nodulectomy in the two referral hospitals. From these, 11 could not be included in the analysis as some essential information was missing. Baseline characteristics of these excluded patients did not show difference from those with we have complete information. Of the21 patients included in the analysis, 16 (76%)were male; and their median age was 30 years(interquartile range (IQR) 22–43 years). The median duration of lymphedema was 15 years (IQR 12–20 years) and the median duration of the presence of nodules was 9years (IQR 7–12). One quarter of the patients (n = 6; 28%) had nodules on both feet. Two thirds of the patients (n = 14; 67%) had stage III lymphedemain one leg and stage II lymphedema in the second leg. The median occurrence of ADLA in the last three months before the surgery was three episodes (IQR 3 to 6) (Table 1).

### Characteristics of surgery and postoperative care

The 21 patients underwent 37 operations: ten patients (48%) had one, six (29%) had two, and five (24%) had three operations. For those who had more than one operation, the median time between two consecutive operations was 29 days (IQR 24–33). The median time of re-epithelization was 23 days (IQR 20–27) (Fig 1). Post-operative wound infection occurred after four out of the 37 procedures. Three months after nodulectomy, two patients reported a single episode of ADLA (each). From the 12 patient who has presented for their follow-up one year

**Table 1. Demographic and clinical characteristics of podoconiosis patient (n = 21) who have undergone surgical nodulectomy in Bahir Dar, Ethiopia between January 2015 and December 2019.**

| Characteristic | Summary |
|---|---|
| Sex | Men:16 (76%) |
| | Women: 5 (24%) |
| Age (years) | Median 30; range 19–76; interquartile range 22–43 |
| Duration of lymphedema (years) | Median 15; range 5–27; interquartile range 12–20 |
| Duration of nodules (years) | Median 9; range 4–20; interquartile range 7–12 |
| Stage | Stage 4 in one foot and stage 3 in the other: 1 (5%) |
| | Stage 3 in both feet: 5 (24%) |
| | Stage 3 in one foot and stage 2 in the other: 14 (67%) |
| | Stage 3 in one foot and stage 1 in the other: 1 (5%) |
| Number of episodes of acute dermato-lymphangio-adenitis in three months before surgery | Median 3; range 0–6; interquartile range 3–6 |

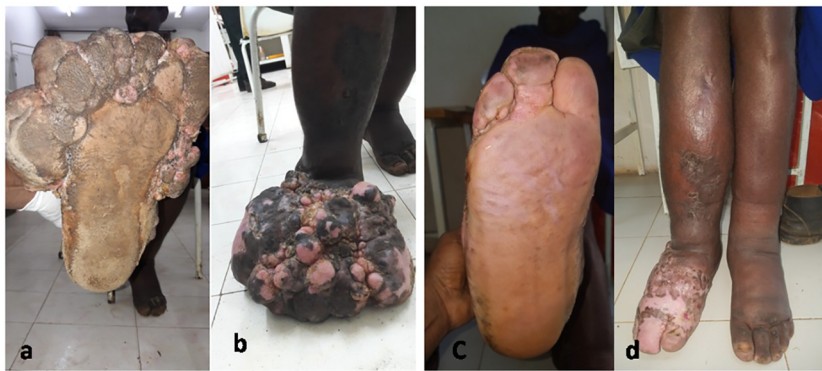

**Fig 1. A 35-year old gentleman with a 15-year history of bilaterallymphedema, and 20 by 17 cm multiple joining nodules on the dorsum of the right foot with lymphorhea, pre-nodulectomy(Fig 1A and 1B); post-nodulectomy (Fig 1C and 1D).**

post-surgery, one patient had mild recurrence of the nodules. In all patients, the podoconiosis stage decreased from stage III to stage II, three months following the last surgery (Table 2).

## Change in total score on Dermatology Life Quality Index over time

The median baseline DLQI before nodulectomy was 16 with an IQR of 13 to 19, (classified as very large effect on patient's quality of life). Two months after surgery, the median score was 11 (IQR 9–14) and six months after surgery, the median score dropped to 5 (IQR 4–7). The total DLQI score was significantly better six months after surgery than before surgery (P<0.0001); the improvement was visible in all six domains of the questionnaire (Table 3 and Fig 2). Also the number of ADLA episodes per three months was significantly lower six months after surgery than before surgery (P<0.0001).

The overall outcome of the procedure was a foot more able to be fitted into a shoe and more practical to wash and care for (Fig 3).

## Wound infection and antibiotic use

In 21(57%) of the surgeries either topical or systemic antibiotic was given as a prophylaxis. And 6 (16%) patient was given oral medication for pain management after the surgery. Only

**Table 2. Characteristics of surgery and postoperative care after nodulectomy (n = 37) for podoconiosis patientin Bahir Dar, Ethiopia between January 2015 and December 2019.**

| Characteristic | Summary |
|---|---|
| Weight of resected nodule (g) | Median 100; range 10–450; interquartile range 50–150 |
| Wound surface after surgery ($cm^2$) | Median 62; range 6–180; interquartile range 38–112 |
| Place of postoperative wound care | Health center: 26 (70%) <br> Hospital: 7 (19%) <br> Self-care at home: 4 (11%) |
| Duration postoperative wound healing (days) | Median 21; range 17–42; interquartile range 20–27 |
| Postoperative pain medication | None: 31 (84%) |
| | Tramadol: 4 (11%) |
| | Diclofenac: 2 (5%) |
| Postoperative wound infection | Yes: 4 (11%) |
| | No: 33 (89%) |

**Table 3. Scores on sections of the Dermatology Life Quality Index before and six months after surgery for podoconiosis patientin Bahir Dar, Ethiopia between January 2015 and December 2019.**

| Section | Maximum score on that section | Before surgery median (IQR) | Six months after surgery median (IQR) | Difference before-after surgery median (IQR) |
|---|---|---|---|---|
| Symptoms and feelings | 6 | 4 (3–5) | 1 (1–2) | 3 (2–4) |
| Daily activities | 6 | 2 (2–4) | 1 (1–2) | 2 (1–2) |
| Leisure | 6 | 3 (2–5) | 1 (1–2) | 2 (2–3) |
| Work and school | 3 | 2 (1–2) | 0 (0–0) | 2 (1–2) |
| Personal relationships | 6 | 4 (2–5) | 1 (1–2) | 2 (1–3) |
| Treatment | 3 | 1 (1–1) | 0 (0–0) | 1 (1–1) |
| Total DLQI score | 30 | 16 (13–19) | 5 (4–7) | 10 (8–12) |

in four (11%) procedures, local wound infections occurred. These infections were diagnosed clinically as there was no access to microbiological investigation. None of the patients developed clinical septicaemia. All wound infections resolved, but there was a clear delay in wound healing: the median time until re-epithelisation time was 21 days in non-infected wounds versus 32.5 days in infected wounds.

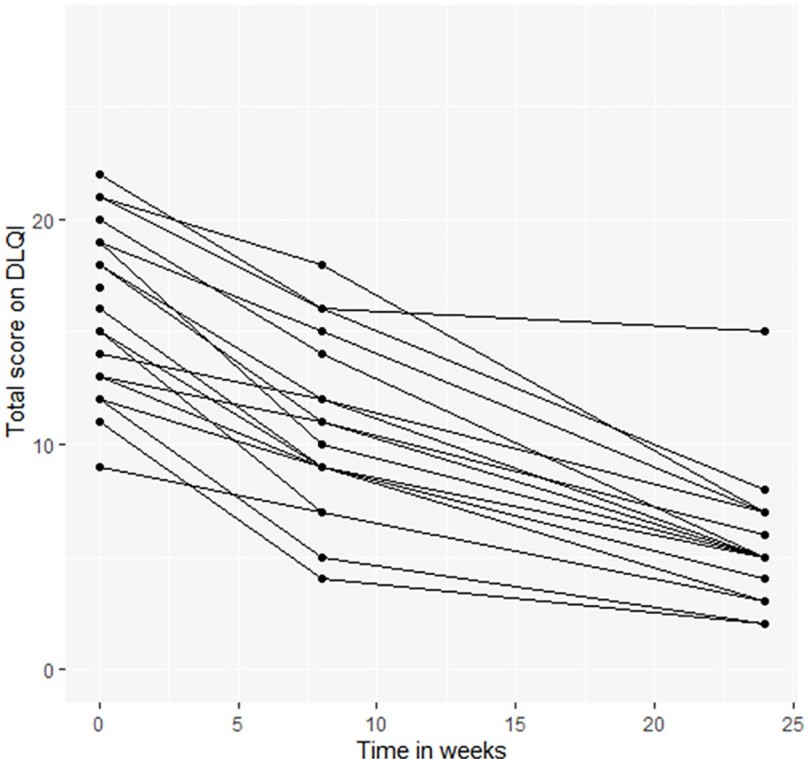

**Fig 2. Change in total score on DQLI over time for podoconiosis patientin Bahir Dar, Ethiopia between January 2015 and December 2019.** Time 0 is before the first surgery. DLQI: Dermatology Life Quality Index.

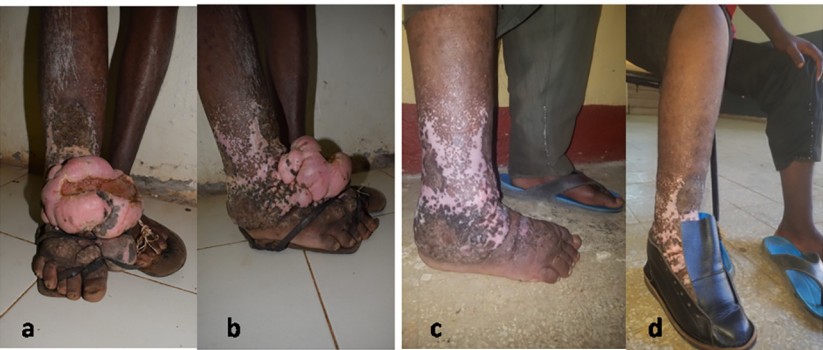

**Fig 3. A 20-years old boy with a 5-year history of right leg lymphedema followed by swelling on the foot with ulceration.** Pre-nodulectomy (Fig 3A and 3B); post-nodulectomy (Fig 3C and 3D).

## Discussion

The findings from this case series reaffirm the safety and effectiveness of surgical nodulectomy in patients with podoconiosis. Authors have written widely on the concept of poor wound healing in lymphedema and various surgical procedures have been proposed to deal significant blood loss, morbidity, infections, permanent disfigurement, and recurrence of symptoms [28,29]. It is assumed that damage due surgery to the lymphatic system impairs the healing of both venous and non-venous wounds[30]. In spite of this assumption, we have demonstrated that surgical nodulectomy can be successful in podoconiosis with acceptable healing rates and an encouraging lack of complications. Most importantly we have seen a significant improvement in patient quality of life following nodulectomy and reduction of ADLA episodes.

Limitations of our study are that we did not exclude filariasis as a cause of edema, lack of an untreated control group and lack of data on the excluded patients. A confounding factor could be that, in addition to the nodulectomy service each patient was advised on daily foot care and use of elastic bandaging which can create a bias in the treatment response. A strength of this study is largest case series reported globally compared to the previous study [14]. Moreover, we also looked the nodulectomy outcomes up to one year after the procedure, including the effect on quality of life and episodes of ADLA.

Rapid wound healing and the absence of major complications is in agreement with our own previous publication[17]. E.W. Price in his 1919's book on podoconiosis also stated that "In cases treated in this way there has been no evidence of re-formation of the nodules" [11]. The large improvement in DLQI score between baseline and six months after surgery indicates that the procedure has a great positive impact on quality of life. The clear reduction of ADLA episodes is also encouraging. In addition to the nodulectomy, lymphedema care and foot care instructions may have attributed to the decrease in ADLA episodes. This finding is also in agreement with the GoLBeT study where simple self-care proofed effective in reducing the frequency and duration of ADLA[13]. Yet, self-care is not a sole option in progressed podoconiosis patient with non-pitting edema and woody-hard fibrous nodules causing an inability to wear foot protection. Also, recurrent disabling bacterial infections are un responsive to self-care [17].

We hope the findings of this study will be used to better understand the role of surgical reduction in the management of lymphedema and give confidence to clinicians to undertake similar life quality enhancing procedures in other settings. We anticipate that these nodulectomy outcomes in patients with lymphedema secondary to podoconiosis may be repeated in other tropical lymphedemas such as secondary to filariasis. This may benefit the patients

directly in the future. The result will also give important information for the Ministry of Health and partners in surgical management of podoconiosis and contribute to improving the National (MOH) podoconiois control guidelines. We recommend future multi-cantered case control studies, with appropriate case definition (e.g. exclusion of filariasis)to validate the current result and address some of the limitations of this study.

In conclusion, this study showed that a simple, resource-appropriate, surgical debulking procedure resulted in significant cosmetic, functional, and self-reported quality of life improvements among people affected with podoconiosis.

## Acknowledgments

We thank all the study participants for their willingness to participate in this study. We also acknowledge staff of the dermatology clinic in FelegeHiwotReferal Hospital and TibebeGhion Specialized Hospital for their assistant in the podoconiosis patient care.

## Author Contributions

**Conceptualization:** Wendemagegn Enbiale.

**Data curation:** Wendemagegn Enbiale, Melesse Gebeyehu.

**Formal analysis:** Kristien Verdonck.

**Funding acquisition:** Henry J. C. de Vries.

**Investigation:** Wendemagegn Enbiale, Melesse Gebeyehu.

**Project administration:** Wendemagegn Enbiale.

**Resources:** Henry J. C. de Vries.

**Supervision:** Henry J. C. de Vries.

**Validation:** Kristien Verdonck, Johan van Griensven, Henry J. C. de Vries.

**Visualization:** Wendemagegn Enbiale, Kristien Verdonck, Johan van Griensven, Henry J. C. de Vries.

**Writing – original draft:** Wendemagegn Enbiale.

**Writing – review & editing:** Kristien Verdonck, Melesse Gebeyehu, Johan van Griensven, Henry J. C. de Vries.

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
