## [Editor Report · Decision Letter 0]

3 Nov 2020

Dear Dr Enbiale,

Thank you very much for submitting your manuscript "Surgical debulking of podoconiosis nodules and its impact on quality of life in Ethiopia" for consideration at PLOS Neglected Tropical Diseases. As with all papers reviewed by the journal, your manuscript was reviewed by members of the editorial board and by several independent reviewers. The reviewers appreciated the attention to an important topic. Based on the reviews, we are likely to accept this manuscript for publication, providing that you modify the manuscript according to the review recommendations. 

Sincerely,

Andrés Felipe Henao-Martínez, M.D.

Deputy Editor

Andrés Henao-Martínez

Deputy Editor

Reviewer #1

This paper has a very good approach related to the assessment of Dermatology quality of live index in patients with surgical treatment including those with a late stage of severity.

I would like to see the inclusion in the diagnosis section of the BinaxNOW filariasis rapid antigen test to rule out LF in podoconiosis, where doubt remains (WHO 2011). Please clarify

Reviewer #2

Authors describe a descriptive analysis of outcomes of nodulectomy in patients with chronic podoconiosis. The results are encouraging and it would prove to be of a high clinical relevance. Few minor corrections can enhance the paper:

*Please recheck your manuscripts throughout for typos and grammar issues. Please use American and not British English (e.g. edema and not oedema). 

*Also in several passages through the text there is a lack of an appropriate space among words including the abstract (see few examples in lines 16, 97, 127, 135, 136, 152, 168, etc.) Please check the entire manuscript carefully.

Methods:

*Please clarify in the method and results section, what other additional interventions were offered to those patients that could have impacted the DLQI scores after surgery? were the patients provided with shoes? Were they given pain or other medications, access to healthcare?

* How this group of patients compared to the most general population? How was your patients selected to the study? 

Discussion:

* Please discuss more limitations of your study. What other unforeseen interventions/biases could have affected the results?

*Please expand and theorize why the intervention worked? Better functionality, less stigma, less progression?

*Please describe and compare your study findings with the natural one year progression of the disease without intervention. How is it different?

*Please propose a follow up study to confirm the findings of this preliminary/descriptive study
---

## [Editor Report · Decision Letter 1]

8 Dec 2020

Dear Dr Enbiale,

We are pleased to inform you that your manuscript 'Surgical debulking of podoconiosis nodules and its impact on quality of life in Ethiopia' has been provisionally accepted for publication in PLOS Neglected Tropical Diseases.

Best regards,

José Antonio Suárez

Guest Editor

Andrés Henao-Martínez

Deputy Editor

---

## [Editor Report · Acceptance letter]

19 Jan 2021

Dear Dr Enbiale,

We are delighted to inform you that your manuscript, "Surgical debulking of podoconiosis nodules and its impact on quality of life in Ethiopia," has been formally accepted for publication in PLOS Neglected Tropical Diseases.

Best regards,

Shaden Kamhawi

co-Editor-in-Chief

Paul Brindley

co-Editor-in-Chief
